

# Rockfall monitoring with a Doppler radar on an active rock slide complex in Brienz/Brinzauls (Switzerland)

Marius Schneider[1,*], Nicolas Oestreicher[1,*], and Simon Loew[1,*]

[1]Department of Earth Sciences, ETH Zurich, Sonneggstrasse 5, 8092, Zurich, Switzerland
[*]These authors contributed equally to this work.

**Correspondence:** Marius Schneider (mariusch@student.ethz.ch)

**Abstract.** We present and analyze a rockfall catalog from an active landslide complex in Brienz/Brinzauls of the Swiss Alps, collected with a new Doppler radar system. This radar system provides a complete and continuous time-series of rockfall events with volumes of $1m^3$ and bigger since 2018 and serves as automatic traffic control for an important cantonal road. In the period between January 2018 and September 2022, 6743 events were detected, which is two orders of magnitude higher
activity than in stable continental cliffs. A few percent of all rockfall events reached the shadow zone, which hosts an important road and agricultural area. The Doppler radar data set allows us to investigate the triggering factors quantitatively. We found that the background rockfall activity is controlled by seasonal climatic triggers. In winter, more rockfalls are observed during thawing periods, whereas in summer the rockfall activity increases with hourly rainfall intensity. We also found that due to the geological setting in an active landslide complex, increased rockfall activity occurs clustered in space and time, triggered
by local displacement hotspots. Thus, monitoring spatial and temporal variations of slope displacement velocity is crucial for detailed rockfall hazard assessment in similar geological settings.

## 1 Introduction

Rockfall is a common hazard in alpine environments, endangering people on roads and in railways due to its high speed and energy (Hungr et al., 2014; Loew et al., 2022; Wyllie, 2015). Rockfalls are distinguished from rock mass falls and rock avalanches
by their physical transport mechanisms (Loew et al., 2022). A common term used for this differentiation is "fragmental" rockfall, where individual blocks mainly interact by impacts with the substrate along the downslope motion, which can be described by rigid body ballistics (e.g., Evans and Hungr, 1993). The spatial characteristics of rockfall source, transport and deposition areas have been described in detail by Evans and Hungr (1993). Of special importance for rockfall hazard analysis and risk mitigation are blocks that have rolled or bounced beyond the base (apex) of talus slopes. In this distal part of the deposition
area, termed shadow, typically only a few boulders are sparsely distributed on the initial substrate.

Many past studies investigated rockfall driving and triggering factors (Matsuoka and Sakai, 1999; Frayssines and Hantz, 2006; Chau et al., 2003; Krautblatter and Moser, 2009; D'Amato et al., 2016; Macciotta et al., 2015, 2017; Peckover and Kerr, 1977). A review of rockfall causal factors has recently been presented by (Loew et al., 2022). Long-term rockfall preparatory causal factors include daily and seasonal thermal cycles driving thermo-elastic strain and subcritical crack propagation



(Gunzburger et al., 2005; Collins and Stock, 2016), brittle rock creep (Cruden, 1970; Brantut et al., 2013), and rock slope deformation in active rockslides (Gschwind and Loew, 2018). Regarding short-term triggers, rockfall activity was found to be related to freeze-thaw cycles in marly limestone cliffs near Grenoble (Frayssines and Hantz, 2006) and in sandstone and shale at Mt. Ainodake, Japan (Matsuoka and Sakai, 1999). Matsuoka and Sakai (1999) report a peak of the rockfall activity during the thawing period. A non-linear response of rockfall to rainfall, with a lower increase of rockfall activity during stronger

rainfall, was often observed (Chau et al., 2003; Krautblatter and Moser, 2009). Both ice warming and thawing are considered important triggering mechanisms (D'Amato et al., 2016; Frayssines and Hantz, 2006), sometimes depending on the season, with high importance of freeze-thaw in early spring (Peckover and Kerr, 1977; Macciotta et al., 2015, 2017). Hence, the major causal factors depend on the location of the study site, its elevation, exposure, and climatic conditions (Macciotta et al., 2017). To understand such triggering mechanisms, a precise evaluation of the time and location of a rockfall event is needed.

A complete rockfall catalog should include a precise starting point with a precise start time, independent of the event time, location and volume. In reality, only a few almost complete rockfall inventories are available. The latter are often based on observations along roads and railways (Peckover and Kerr, 1977; Macciotta et al., 2015; AWN, 2022). In the case of the Canadian Cordillera, railway personnel recorded each rockfall terminating on or close to the railway in a catalog. Similar methods are used by road authorities in the Swiss Alps. Since it is known that the rockfall reach angle highly depends on its

mass (e.g., Copons et al., 2009), such catalogs include a bias towards large rockfall events and only a rough estimation of the event timing is possible. Other methods to collect rockfall data include tree-ring injury analysis (Stoffel and Bollschweiler, 2008), rockfall volume estimations with traps (Krautblatter and Moser, 2009; Sass, 2005), time-lapse cameras (D'Amato et al., 2016; Frayssines and Hantz, 2006), or simple audience reports (AWN, 2017). All of these methods lack either temporal or spatial completeness and precision. More advanced techniques to detect rockfalls are repeated laser scans of rock walls

(Rabatel et al., 2008), which can narrow down the volume and timing of events. However, the release of several small events from the same area between two acquisitions cannot be distinguished, among other biases (Dussauge-Peisser et al., 2002).

In this study we apply a new method of rockfall monitoring which was developed by Geopraevent, using a Doppler radar for very fast movement detection and immediate and automatic trigger of response, providing detailed temporal information for each rockfall event in all-weather conditions and darkness (Geopraevent, 2019). This contribution aims to analyze rockfall

frequency in time and space on the active rock slide complex of Brienz/Brinzauls to understand drivers and event triggers in such environments. We use the Doppler radar technology to monitor rockfall time-series with unprecedented completeness, combined with robotic total station and ground-based interferometric radar measurements of the moving landslide. We elaborate on the benefits of the advanced Doppler radar monitoring system and show its potential and limitations for generating a complete rockfall catalog and immediate alarm. This is especially important for elements at risk located in the rockfall shadow

area. Therefore this study also provides new insights into statistical properties of rockfall blocks reaching this area.



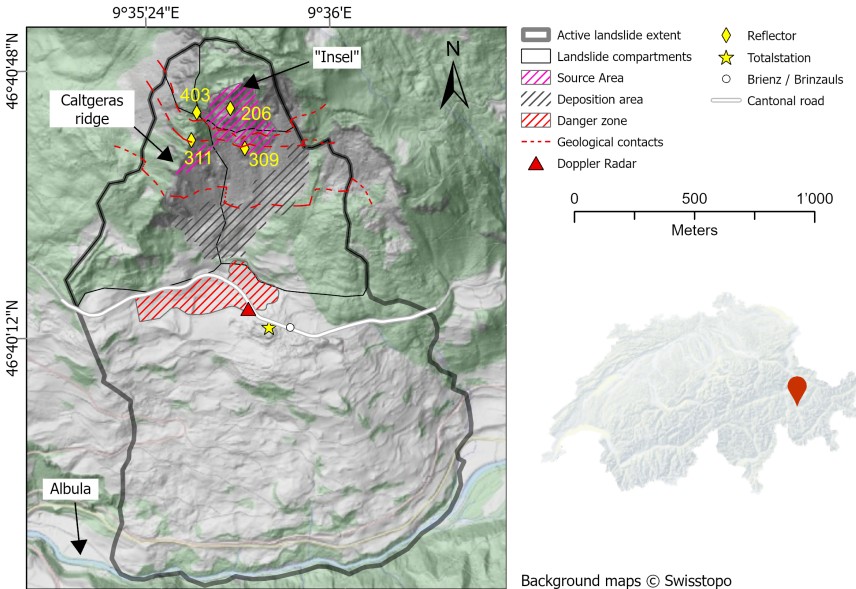

**Figure 1.** Overview map of the study area. The thick black line indicates the extent of the active landslide complex, limited in the south by the Albula river. The thin black lines show landslide compartments (simplified after BTG (2022b)). The magenta, gray and red hatched areas show the source, deposition and danger zones, respectively. Red lines mark important formation boundaries between Flysch, Allgäu-Formation, Raibler Formation, and Vallatscha Formation, from bottom up after BTG (2022b). Background colored in green indicates forested area.

## 2 Study Site and Geological Setting

The study site of this project is located above the village of Brienz/Brinzauls (1150 m. a.s.l.), in the canton Grison of Switzerland. Here, an old deep-seated mountain slope deformation hosts a very active landslide complex with a volume of $170km^3$ and annual displacements of 1 to 10m into southern directions (BTG, 2022a). The active landslide complex extends from an

altitude of 1800m a.s.l down to the Albula river at about 800m a.s.l.. Tectonically, the Brienz mountain slope deformation is located in the transition area between the Penninic and Eastern-Alpine sedimentary nappes (Brauchli, 1921). The landslide consists of Flysch and Allgäu formation (Fm.), composed of schists with alternating limestone/sandstone and marl layers, overlaid by Raibler Fm. (Rauwacke, Dolomite) and Vallatscha Fm. (Arlberg Dolomite) (BTG, 2022a). The Vallatscha Fm. hosts most of the rockfall source areas and dips steeply in downslope direction. Other source areas are located further downslope

in the Raibler and Allgäu Fm. of the Caltgeras ridge, which dip subhorizontally or slightly into the slope. The groundwater table in the rockfall source is more than 100 m deep due to the high permeability of the fractured and karstified dolomites and the topographic setting. Different kinematics are observed within the active rock slide, and the landslide complex is divided



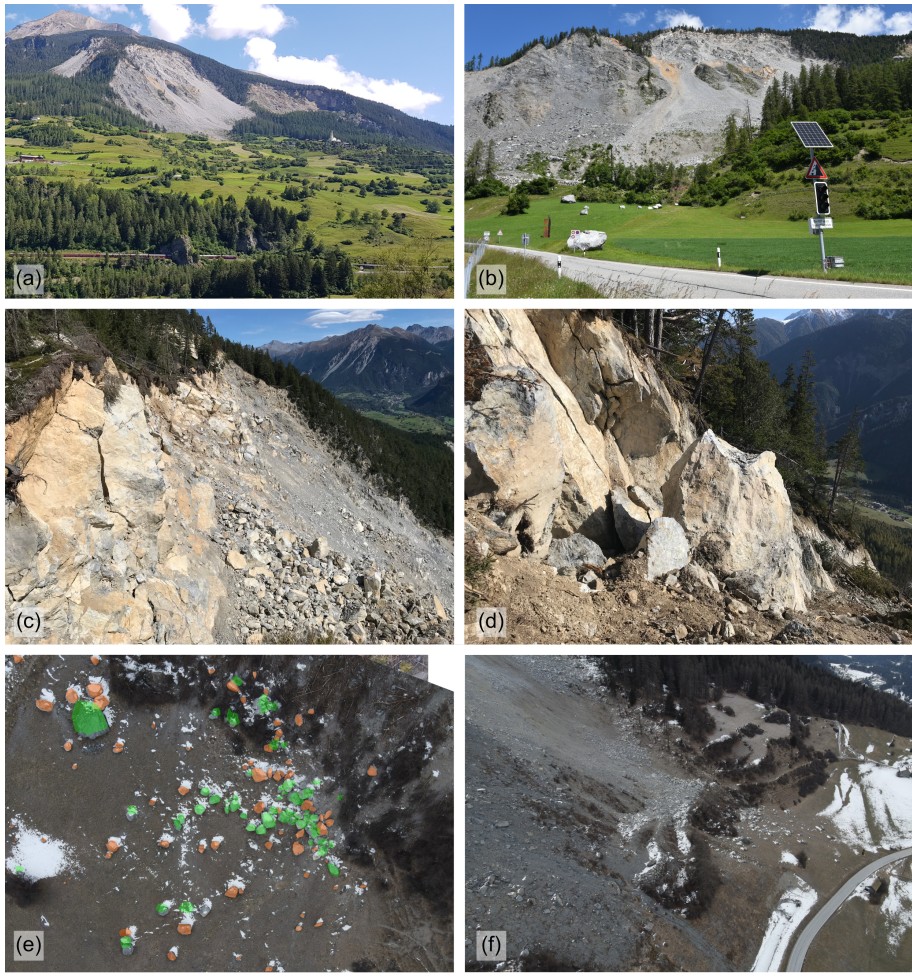

**Figure 2.** Photos from the study site. **(a)** View from Ruteira (above Tiefencastel) on the complete mountain slope deformation of Brienz/Brinzauls, looking NNE. **(b)** View on the rockfall release, transport and deposition areas from the Brienz village and cantonal road, looking north. The danger zone with meadow and cantonal road is in the foreground, trees on the rockfall dam are located between the danger zone and talus cones. **(c) and (d)** Upper sector of release area showing fractured Arlberg dolomite, looking east. **(e)** Deposited blocks on the grassland in the shadow zone. Green blocks were deposited before 21 April 2022. Orange-colored blocks were deposited in the period between 21 April and 30 January 2023. **(f)** Side view of the partially filled rockfall dam and the adjacent shadow zone.

into different compartments (Loew, 2022). This results in differential displacements within the rockslide complex, making monitoring indispensable.

The active slope instability was under scrutiny for decades, when moderate accelerations took place in 2000-2001 (Sartorius et al., 2022; BTG, 2022a). In 2009, systematic GPS measurements started. Since 2011, a robotic total station monitoring system monitors about 30 reflectors. In 2012, a dam was built above the cantonal road to retain blocks reaching the danger zone. In 2019 ground-based interferometric synthetic aperture radar (GBInSAR) was installed, together with a Doppler radar rockfall





**Table 1.** Used meteorological data sets. Data available on meteoschweiz.admin.ch

| Station | Initials | Elevation | Parameter | Interval | Distance |
|---------|----------|-----------|-----------|----------|----------|
| Brienz/Brinzauls | BRINZ | 1758 (m a.s.l) | Temperature ($°C$) | 1h | local |
| Tiefencastel | TIC PM | 899 (m a.s.l) | Precipitation ($mm d^{-1}$) | 24h | 1.7km |
| Alvaneu | ALV PM | 1162 (m a.s.l) | Precipitation ($mm d^{-1}$) | 24h | 4.2km |
| Alvaneu | ALV SM | 1162 (m a.s.l) | Snow height ($cm$) | 24h | 4.2km |
| Savognin | SVG TA | 1172 (m a.s.l) | Temperature ($°C$) | 1h | 8.1km |
| Savognin | SVG PA | 1172(m a.s.l) | Precipitation ($mm h^{-1}$) | 1h | 8.1km |

monitoring system for automatic closure of a cantonal road during rockfall events (Fig. 1). Rock slide acceleration mostly takes

place during winter and spring, when snowmelt occurs (Fig. 3). During summer months and fall, landslide velocity fluctuates

but mostly shows no trend to acceleration (Fig. 3).

Several larger rockfall events were observed in recent years. Blocks up to $60 m^3$ reached the cantonal road in 2015 and 2019 while smaller blocks regularly come to hold in the adjoining agricultural grassland. We define this grassland and the immediate surroundings as a hazard zone (Fig. 1), since rockfalls reaching into this area are of special interest due to the increased damage

potential (traffic on cantonal road). However, our defined hazard zone is not related to the hazard zone defined by the canton. Rock mass falls with volumes up to $100000 m^3$ took place in 2001 and 2015 where blocks were able to pass the retaining dam on the side and reach far into the shadow zone.

## 3 Data and Data Processing

### 3.1 Meteorological Data

For this study, meteorological data sets from the MeteoSwiss stations Tiefencastel, Alvaneu, and Savognin and local station above the rockfall release area from the Swiss Seismological Survey are used (Tab. 1). Tiefencastel and Alvanaeu are manual measuring stations for precipitation and snow height, respectively. These measurements are conducted every day at 7.30 AM. In Savognin, hourly precipitation and temperature values are measured.

The temperature sensor at the local weather station is situated $2m$ above the ground, is south-facing with partial exposure

to sunlight and is equipped with a radiation shield. However, we observe that during summer, the temperature in Savognin is about $1°C$ higher, but in winter about $2°C$ lower compared to the local weather station (BRINZ). This phenomenon is likely caused by topographical effects, which capture cold air mass in the valley bottom during winter and does not allow a reasonable temperature gradient calculation. However, we further compared the temperature data sets from Savognin and Brienz/Brinzauls and found an overall correlation of 0.93 between both stations. Due to this high correlation, we use the temperature data set

from the local Station of Brienz/Brinzauls for temperature analysis on release area elevation.



To assess the influence of temperature on rockfall activity, we used two different metrics to investigate potential causal factors. First, we sum the hours with positive temperatures for each day with the temperature data set from Brienz/Brinzauls (positive degree hours). Secondly, freeze-thaw (FT) cycles can be divided into three phases: negative cooling when the temperature is below $0°C$ and temperature decreases, negative warming when temperature is increasing but still below $0°C$ and

thawing when temperature is above $0°C$. Assuming constant water seepage into the rock mass during winter, ice production can be characterized by the freezing potential (FP) (Matsuoka, 1994; D'Amato et al., 2016). The FP for one single FT cycle is defined by

$$\text{If: } \int_{t_0}^{t} (Tf - T(t))dt < 0, \text{ then } FP = 0 \tag{1}$$

$$\text{If: } \int_{t_0}^{t} (Tf - T(t))dt > 0, \text{ then } FP = \int_{t_0}^{t} (Tf - T(t))dt \tag{2}$$

where,

$$t_0 = \text{Start of freeze-thaw period} \tag{3}$$

$$Tf = \text{Freezing temperature of water } (0°C) \tag{4}$$

$$T(t) = \text{Temperature at time t} \tag{5}$$

The modelled FP allows us to identify all three phases during an FT cycle. The frost phase begins as soon as the temperature

is below the freezing point of water and ends as soon as melting occurs. This is the case when temperature is above $0°C$ again. However, during freezing, both negative cooling and warming can take place. Melting happens when the temperature is above the freezing point of water. It will continue until the FP reaches zero, and no more ice is available (D'Amato et al., 2016). We calculate the FP with the elevation-corrected daily mean temperature data set from Station Brienz/Brinzauls from October 2018 until November 2022. We defined each phase according the following conditions:

$$\text{Negative cooling} = T(t) < 0 \text{ and } dT/dt < 0 \tag{6}$$

$$\text{Negative warming} = T(t) < 0 \text{ and } dT/dt > 0 \tag{7}$$

$$\text{Thawing} = T(t) < 0 \text{ and } FP(t) > 0 \tag{8}$$

The input of water at the soil surface from snowmelt and rain (Surface Water Input, SWI) in the area of rockfall release was physically modeled based on local snow height measurements by SLF (2022) and used in this study. The SWI highlights the

importance of the snowmelt on water available for infiltration to groundwater in the study area. We consider the SWI a good indicator of the state of soil wetness in the shallow subsurface in our study area. However, we note that the groundwater recharge can strongly differ from the SWI, mainly due to evapotranspiration and surface runoff. The mean precipitation intensity per day



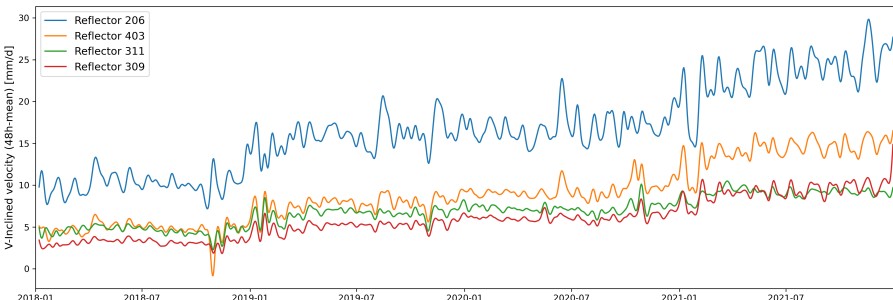

**Figure 3.** TPS-derived landslide velocity (low-pass filtered with a cutoff of 1/14 per day) in different compartments along line-of-site recorded from a total station located in Brienz village. TPS reflector positions are indicated in Fig.1. Velocities show a long-term trend to higher velocity, superimposed by seasonal accelerations in wintertime.

was calculated using Savognin station. We counted the number of hours with precipitation greater than zero. The total amount of precipitation per day was then divided by the number of hours of precipitation, resulting in the daily mean precipitation

intensity.

## 3.2    Doppler Radar

Radar is a general abbreviation for RAdio Detecting And Ranging and is exploited in the microwave range of the electromagnetic spectrum (e.g., Cavell, 2017). By emitting radar waves with a known frequency from a stationary location on a moving object, the reflected radar waves show a different frequency. This phenomenon is known as the Doppler effect (Doppler, 1903)

and allows the detection of relative velocity and position of moving objects in the field of view. The corresponding change in the frequency is known as the Doppler shift frequency. This allows to detect and track fast movements, which is used in regular daylife.

The implementation of a Doppler radar for monitoring rockfalls was used for the first time in Brienz/Brinzauls. Originally, this technology was used to monitor snow avalanche slopes in the Valais Alps. Since January 2018, the Doppler radar system is

fixed to the wall of the old school building in Brienz/Brinzauls, monitoring the active rockfall slope continuously. The Doppler radar operates at a frequency of $10\,GHz$, where damping due to snow and rain is relatively low (Gassner et al., 2022). The device has a 90° horizontal field of view into 347° North direction. The highest radar sensitivity is set at 20° vertical elevation. The monitored area covers about $0.5km^2$ with a maximum distance of approximately $1km$. Measurements are done with a spatial resolution of $20m$ in a 347° N inclined coordinate system. The coordinate system origin is set to the Doppler radar

position. The minimal volume required to detect moving objects depends on the distance to the radar system: within a distance of 100 m the radar can detect moving masses larger than $0.1m^3$. Increasing the distance to $1km$, a minimum volume of $1m^3$ is necessary for detection. Further, the Doppler radar in Brienz/Brinzauls is only able to detect movements with a velocity in the range of meters per second (Gassner et al., 2022). Slower movements are not detected. The system is combined with a triggered



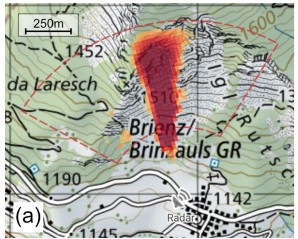 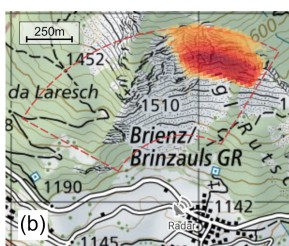 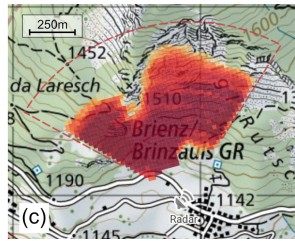

**Figure 4.** Screenshots of rockfall activity maps provided in the Geopraevent data portal, background maps from Swisstopo. **(a)** Normal event pattern with a wider spread in the head region, getting narrow in the deposition area **(b)** East event pattern **(c)** TPWE event with an indistinctive pattern, caused by an incoming weather front.

optical camera, providing 15-20 high-quality images of each rockfall event at 3-second intervals. These images indicate that
many large rockfall events are rock mass failures creating dozens of fragmental rockfall blocks in the transit area.

As the Doppler radar continuously scans the rockfall slope, permanent calculations of the signal-to-noise ratio are going on. Therefore, the mean activity of the last 30 minutes is calculated. Then, the signal is set in ratio to the noise. If this exceeds the threshold, a potential rockfall event is identified. Advanced algorithms further validate the detected potential rockfall and check if the pattern is related to other sources (e.g., wildlife, UAV and others). This validation leads to a very low false positive
(FP) rate of about 1% and is processed within seconds (Gassner et al., 2022).

### 3.3 Rockfall data processing

The Doppler radar of Brienz/Brinzauls creates a unique data set for rockfall information in time and space. For each event, an exact timestamp of start and end time is registered. With these timestamps, an average frontal speed is estimated. The radar data also contains spatial information about the rockfall event. For this purpose, the activity of each point within the coordinate
system is analyzed. The system detected 6743 single events (time stamps) from 8 January 2018 to 5 October 2022.

The provided activity map (Fig. 4) is used to analyze the source, transit, and deposition areas. The activity map provides the averaged signal-to-noise ratio of the radar amplitudes for every grid cell for the time period of the entire rockfall motion. The time-lapse camera images indicate that for multi-block or rock mass fall events, the activity map mainly describes the tracks of the larger blocks with the main intensity density. Therefore, the activity area is wider at the start and narrower with travel
distance (Fig. 4a). Due to topographic effects, events released from the east region of the rockfall slope show a wide activity pattern regardless of the actual extent (Fig. 4b). Therefore, start points from the east area need to be interpreted cautiously. Several other events show an indistinct scatter on the activity map (Fig. 4c). Incoming weather fronts (thunderstorms, hail, snowfall) during a rockfall event led to low signal-to-noise ratio activity maps. Therefore, these events are not classified as false positives but as true positive wrong extent (TPWE) events. Since the spatial information of TPWE is inaccurate, they
need to be removed from the data set for the spatial analysis but are kept for the temporal analysis.

TPWE events are identified and removed from the spatial analysis data set with a supervised machine learning (ML) method (Abadi et al., 2015; Renotte, 2022). 1000 rockfall activity maps were randomly picked and classified manually for training and





validation purposes. The manual catalog contains 29 TPWE and 971 normal events. This data set is further split into training set 70%, validation set 20% and test set 10%. The ML is trained by choosing random batches of rockfall activity maps from the training set. One training epoch contains 22 batches with 32 images. We trained the model for 120 epochs with a test and validation loss of 2.0e-5 and 1.3e-5, respectively. With this method, 185 TPWE (3.1%) are detected and removed from the spatial events catalog (185 events for the spatial analysis).

The following assumptions are made to calculate the starting point. First, rockfall is a gravitational process and can only happen from top to bottom (decreasing north). Second, the event always starts at the highest point of the activity map (farthest distance from the radar). Thus, y is the maximum north value of the event activity map. However, we must consider the lateral variability in the east-west direction (Fig. 4a). For this purpose, we calculate the mean of the corresponding x values for maximum north up to maximum north minus 35 meters. It should be noted that the accuracy of the radar in the east of the rockfall area is low (Fig. 4b). Therefore, the starting points in this area should be considered with caution. Although we have developed an elaborate procedure to remove TPWE events from the data set, some remain unnoticed. The start point analysis enabled us to identify such undetected TPWE events with an unrealistic extent (start point outside the radar field of view), which are manually removed from the spatial data set using Arc GIS Pro.

## 3.4 Aerial Image Analysis

To investigate the time history and rock fall deposition frequency in the danger we have analysed a series of high-resolution (10 cm ground resolution and higher) aerial images. The study employed swissimage-orthophotos obtained from swisstopo for the years 2003, 2006, 2009, 2012, 2015, 2019, and 2020. Furthermore, two drone flights were conducted using a commercial DJI Mavic Air 2 drone in April 2022 and January 2023. Subsequently, each image was carefully compared with the preceding one, and the blocks that were newly deposited were identified and marked manually using ArcGIS.

The accuracy of the method utilized for detecting and describing the number of blocks deposited in the hazard zone is constrained by several factors. The prolonged interval between successive orthophotos obtained from Swisstopo contributes to increased uncertainty. This is attributed to the removal of boulders by farmers from the grassland during the period of low rockfall activity before the year 2020. Moreover, the limited image resolution restricts the detection of rocks of a certain minimum size. Furthermore, the method cannot differentiate between the deposition of a single block or multiple rocks during an event.

Despite these limitations, it is crucial to document the progression of the number of blocks deposited in the hazard zone and any variations that may arise. However, it is anticipated that the quantitative analysis, especially for the period preceding 2020, may be underestimated, and therefore, must be interpreted cautiously.





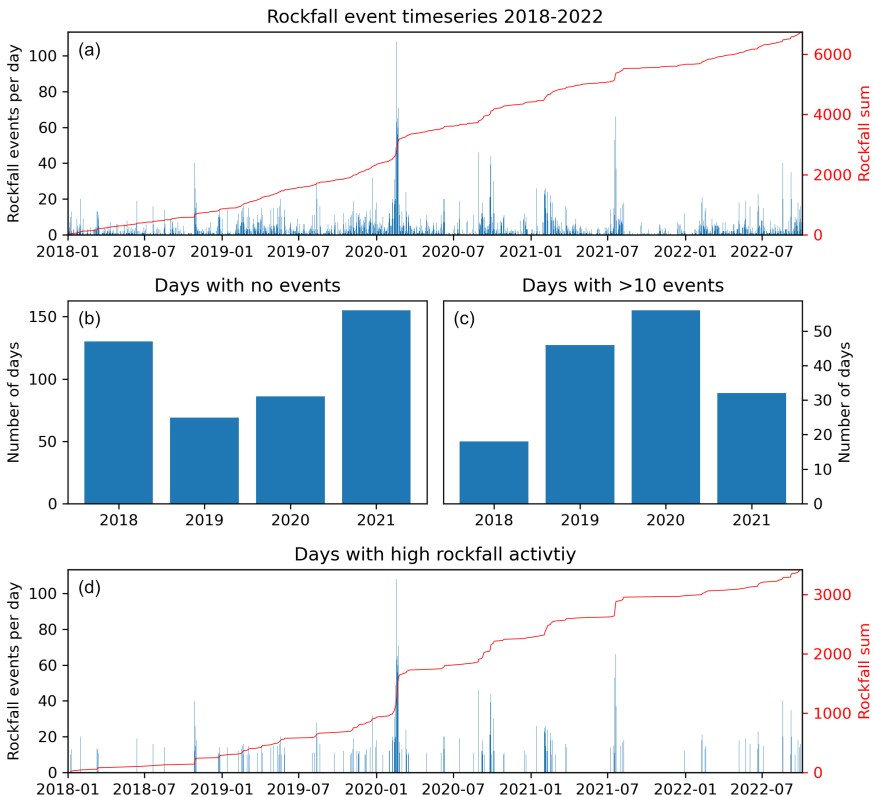

**Figure 5.** Rockfall event time-series from January 2018 to October 2022. **(a)** Rockfall event count **(b)** Days with 0 events during the sampling years 2018-2021**(c)** Days with more than 10 events during the sampling years 2018-2021 **(d)** Time-series with high rockfall activity (more than 10 events registered)

## 4    Results

### 4.1    Long-term rockfall variability in time and space

From January 2018 to December 2021, 5667 events were detected from the Doppler radar. In general, rockfall events in
Brienz/Brinzauls show a large variety in fragmentation, velocity and event volume. This is related to the geological complexity
of the source area, composed of different lithologies, weathering grades, fracturing and strength. Rockfalls released from
the main rockfall scarp, where predominantly massive dolomite is present, are mostly multi-block failures from bedrock and
show the lowest grade of fragmentation. Accordingly, blocks with volumes up to $60m^3$ can reach the danger zone by rolling,
bouncing and sliding. During the observation period, all large events which reached the grassland were released from the main
scarp. Rockfalls from the Caltgeras ridge, where lower strength graywacke with dolomite or even Allgäu schists is present,
show a high grade of fragmentation. Rockfall events from this source area mainly propagate by sliding and falling. If dolomite





blocks are present, jumping and rolling may also occur, but limited to single blocks and therefore reaching further than the fragmented Allgäu schists. In either case, the event volume is in the range between 1 to $100m^3$, but with a much higher frequency of small events volumes ($1m^3$). We only rarely observed events released from colluvial deposits. However, after the

failure of a larger rock mass (i.e. February 2020), some later rockfall events were released from the colluvial slope, mobilizing only very limited volumes. This is reflected in a small second peak in the N-S-histogram of Fig. 6.

The number of rockfalls per day fluctuates strongly, with a mean of 3.8 events per day. Also the number of rockfalls per year varies significantly, with 856 (2018), 1467 (2019), 2088 (2020), and 1256 (2021) events. In 2022, 1076 rockfall events were detected until October 5. This corresponds to a mean of 3.9 events per day. Two periods with intense rockfall activity occurred

in February 2020 and July-August 2021, respectively, with 108 and 78 events recorded on a single day. A slight long-term increase in rockfall activity was present during 2019, ending with the rockfall cluster in February 2020. Since January 2022, a long-term increase in rockfall activity is present, which can be seen in Fig. 5a. In total, 439 days did not have any rockfall events detected. 152 days exceeded the 90th percentile of rockfall activity (equal or over 10 events in a day), hereafter considered as days with high rockfall activity. 18 took place in 2018, 46 in 2019, 56 in 2020, and 32 in 2021 (Fig. 5c). The two years on

record with the largest number of high rockfall activity days are also the ones with the least number of days without any event recorded. Conversely, there were many more days without events in years with a lower number of high rockfall activity days (e.g., 130 in 2018, 69 in 2019, 86 in 2020 and 155 in 2021, Fig. 5b).

During the period in record, a clustering of days with high rockfall activity can be identified in Fig. 5d, for example in February 2020 and 2021, or August 2021. At these times, the number of events usually first increases exponentially and

rapidly decays back to the baseline activity after the peak is reached. Such cluster patterns can also be observed in the rockfall frequency plot of Fig. 5a.

We performed a spatial analysis to find patterns in rockfall events' spatial origin. A change in rockfall starting locations would influence the lithology of the involved blocks, inform about transport processes, and modify the rockfall hazard for the road and danger zone at the bottom of the slope. Spatial radar data are only available since November 3, 2018; hence we

decide not to include 2018 in the analysis. During this period the average rockfall starting location has moved westwards (about $40m$) from 2019 to 2021(Fig. 6a). Significantly fewer rockfall events have originated from the easternmost part of the slope since 2020. Simultaneously, the average rockfall release location has moved southwards (hence downwards), with a marked change between 2019 and 2020. Overall, most rockfall events are released from a compartment (termed "Insel", see Fig. 1) in the Vallatscha Fm., especially at the contact of Vallatscha and Raibler Fm. on the west side of the slope. Some TPWE events

remain in the data set, where the release area is wrongly attributed to a part of the slope out of the radar field of view. Fig. 2c and d show detailed photographs of the main release area in the massive dolomites of the Vallatscha Fm.

In the upper part of the rockfall slope the transport paths are dominated by bedrock outcrops (Fig. 2) and are strongly channelized. The talus cones begin at about one third of the rockfall slope elevation and end at the retaining dam. Most blocks are deposited on these talus cones below the release areas (Fig. 2b). We observe a size grading described by increasing block

size with increasing distance from the source area. Although the dam can hold back a large part of the blocks, from time to time, larger blocks get onto the grassland adjoining the cantonal road. In the past, large blocks (mainly Arlberg dolomite) were



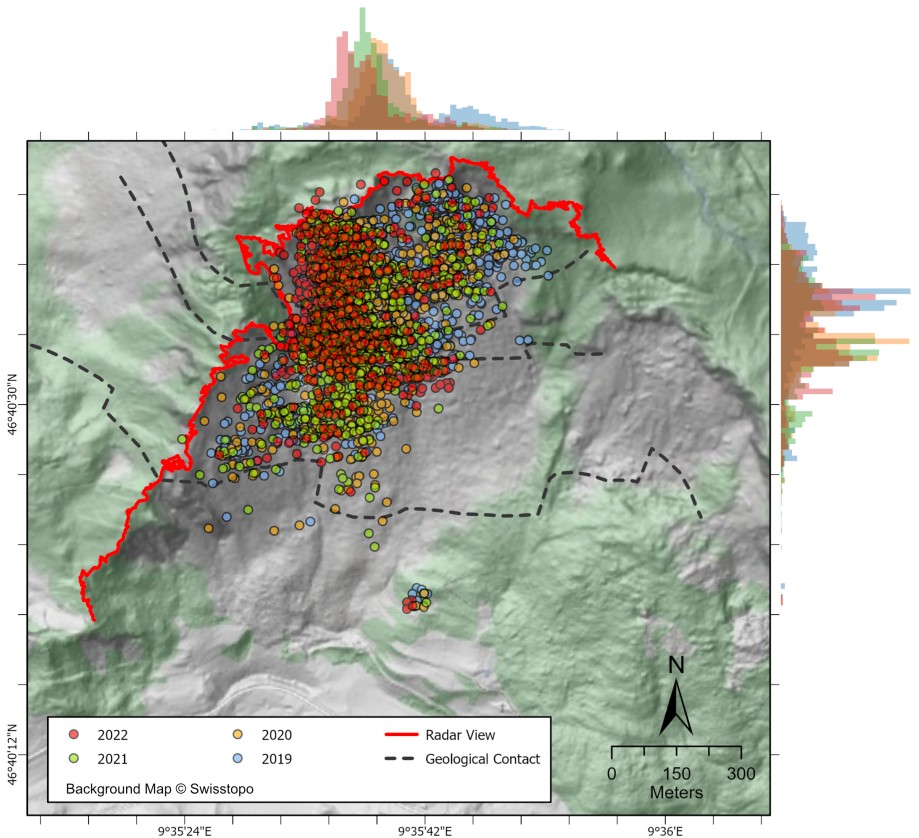

**Figure 6.** Calculated event start points derived by year (2019-2022). Due to low spatial data availability, 2018 is not included. The coordinate system is WGS 84. The histograms show the probability density of events in N-S/E-W for each year. The red line indicates the maximum radar view (crest line). The dashed black lines are geological boundaries after BTG (2022b). Green background indicates forested area.

even able to pass the cantonal road. Aerial images recorded between 2003 and 2022 have been analyzed for rockfall deposits in the danger zone corresponding roughly to the rockfall shadow area (Fig. 1 and 7). We found that in total 183 blocks reached this shadow area during this long monitoring period, having a travel angle ranging between 25 and 27°. However, it has to

be mentioned that this number tends to be highly underestimated, since the time gap between each Swisstopo orthophoto is large and land users usually removed the blocks quickly after deposition until 2021. From a magnitude-cumulative frequency analysis after Hungr et al. (1999), we can derive a long-term return period for the large 10 and $100m^3$ size blocks reaching the danger zone of 1 and 8 years, respectively. Flying rock fragments have been observed but not recorded systematically.

In the shorter time period between 2018 to 2022 even 166 blocks reached the grassland or shadow area. In addition, we

observed that 88 blocks were deposited here between April and December 2022. The frequency of blocks reaching the shadow zone is thus significantly increased compared to the previous periods. Assuming that a single block is attributed to an event, this corresponds to a rate of 6% of all events between April and December 2022 that enter the shadow zone. Over the whole



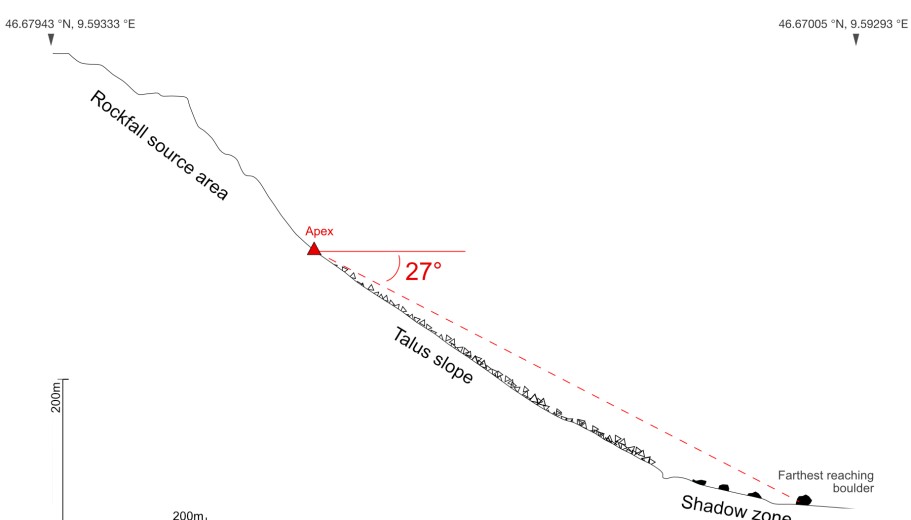

**Figure 7.** Profile through the rockfall slope of Brienz/Brinzauls showing the rockfall source area (schematic), the talus cone extend and the farthest reaching boulder (approx. $60m^3$) in the shadow zone (Event dated 13 August 2019 14:03 local time).

sampling time, about 2.5% reach the shadow area. We examined the time-laps camera images of events reaching the shadow zone which showed that the blocks trajectory primarily passed the dam on the east side. This pattern is also evident in the
rockfall event map of the canton Grison (AWN, 2022). However, since 2020 a higher frequency of blocks traveling directly over the dam by rolling and bouncing was observed. Additionally, the size distribution of the blocks has also changed, with smaller blocks ($< 1m^3$) being deposited in the shadow zone after 2020. Before, mainly larger blocks ($> 1m^3$) were able to pass the dam.

## 4.2 The February 2020 Event Cluster

We have identified clustering rockfall events as a typical characteristic of rockfalls released from active rock slope movements. Here we present in detail the rockfall events cluster of February 2020 because it corresponds to the highest rockfall rate observed in the monitoring period. From January 1 to February 29, 878 events were registered, corresponding to 13% of the total number of events registered in the entire rockfall catalog. The surface water input from snowmelt and rainfall was not exceptionally high during this period (Fig. 8). The temperature was fluctuating around the freezing point, as is usual for this
time of the year (Fig. 8), and was progressively warming up, in particular after the end of a cloudy period, around March 15, 2020 (Fig. 8). While the cluster of rockfall events cannot be easily explained by weather or hydrologic factors, none of the nearby reflectors or GNSS stations recorded a significant increase in the velocity of the corresponding landslide compartment (Fig. 1).

On the other hand, detailed inspection of the ground-based InSAR data and optical images shows that a small ($600m^2$)
patch of this compartment was moving at a much higher velocity than the rest of the landslide. In January 2020, this rock



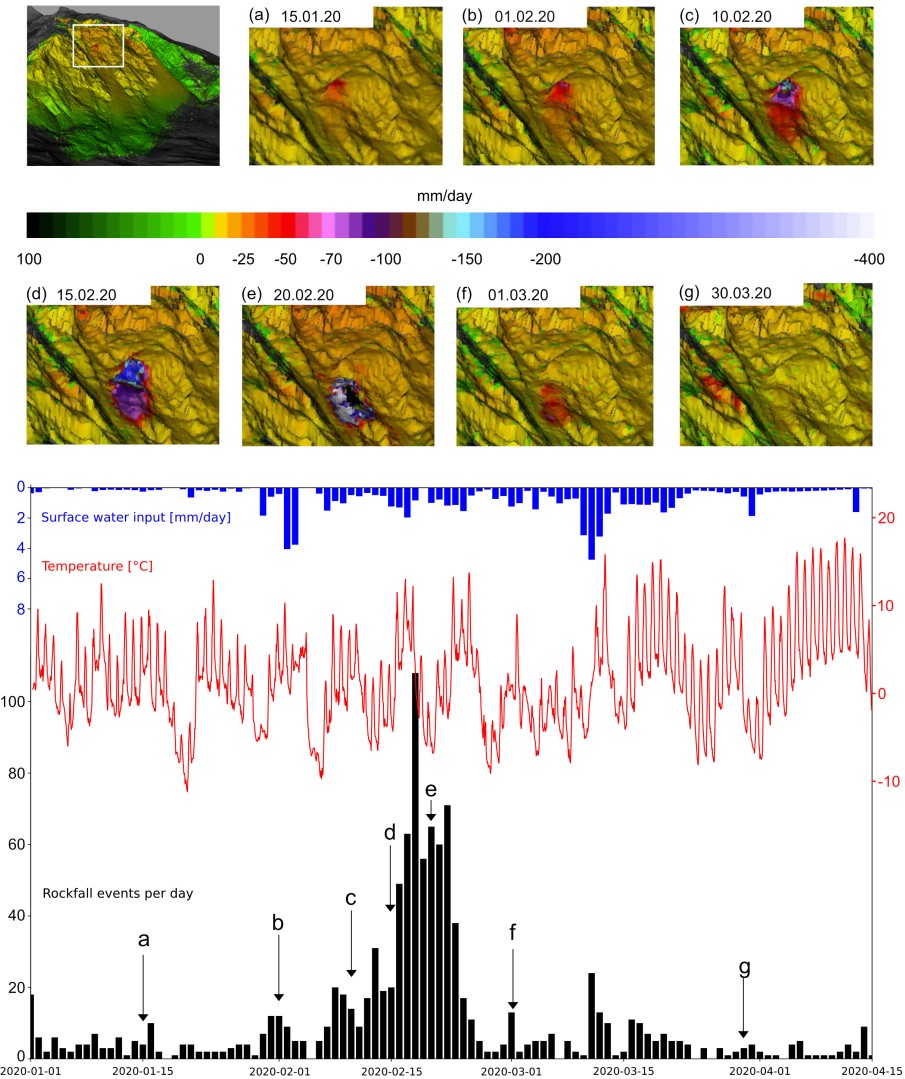

**Figure 8.** Rockfall event cluster from February 2020. Top: ground-based interferometric synthetic aperture radar (GBInSAR) images from January 15, 2020 to March 30, 2020 for the landslide displacement. Bottom: time-series with surface water input from SLF (2022) (blue bar), temperature from Brienz/Brinzauls (red line), and rockfall events per day (black bars). Letters and arrows indicate the corresponding GBInSAR image acquisition time.

mass block started to accelerate and reached a velocity peak of $1.5 md^{-1}$ around mid-February 2020 (see Fig. 8a to e), when it collapsed. The rockfall release points were grouped in the area where the local acceleration took place (Fig 9). On February 18, the highest rockfall frequency was recorded, with about 100 individual rockfall events. Many of them were released from the debris cone, i.e. the deposits from the collapsed rock mass patch were reactivated (Fig 9). Afterward, the rock mass velocity decreased again (see Fig. 8f), back to the velocity of the surrounding parts of the slope (0.5 to $5 cmd^{-1}$) at the end of March




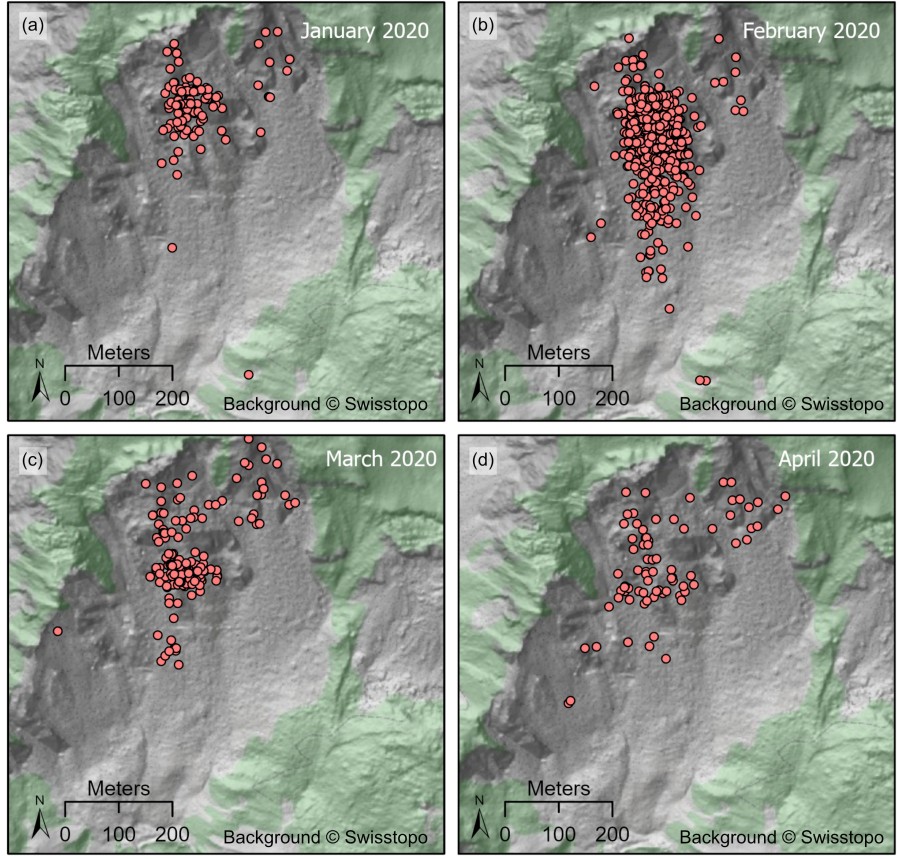

**Figure 9.** Rockfall release points in **(a)** January, **(b)** February, **(c)** March and **(d)** April 2020. Green background indicates forested area.

(see Fig. 8d). During this period, the rockfall release points were more scattered across the slope, compared to the Months January and February 2020 (Fig. 9).

## 4.3 Weather as a Rockfall Trigger

The Doppler radar data set of the Brienz/Brinzauls rockfalls was also investigated for potential climatic triggering factors, such as surface water input from rainfall and snowmelt, temperature, and freeze-thaw cycles. Although the SWI data set from SLF (2022) is limited to summer 2021, we used it as the best approximation for meltwater availability. We used Pearson's correlation factors to investigate linear correlations between climatic factors and rockfall frequency. At Brienz/Brinzauls, the potential water input is mainly dominated by snowmelt during winter, while in summer, strong thunderstorms lead to large amounts of available water input in a short time. Because of the significant seasonality in temperature and surface water input, we divide the data set into summer (April to September) and winter (October to March) months. This allows a more precise analysis of the meteorological influences on rockfall activity.



**Table 2.** Pearson's correlation coefficients between daily rockfall activity and daily weather variables. Precipitation data from Tiefencastel, Temperature from Brienz/Brinzauls, SWI from SLF (2022).

| Period | SWI | Precipitation | Average daily temperature |
|---|---|---|---|
| Winter | 0.23 | 0.15 | 0.10 |
| Summer | 0.23 | 0.33 | -0.13 |
| All rockfalls | 0.13 | 0.23 | -0.05 |

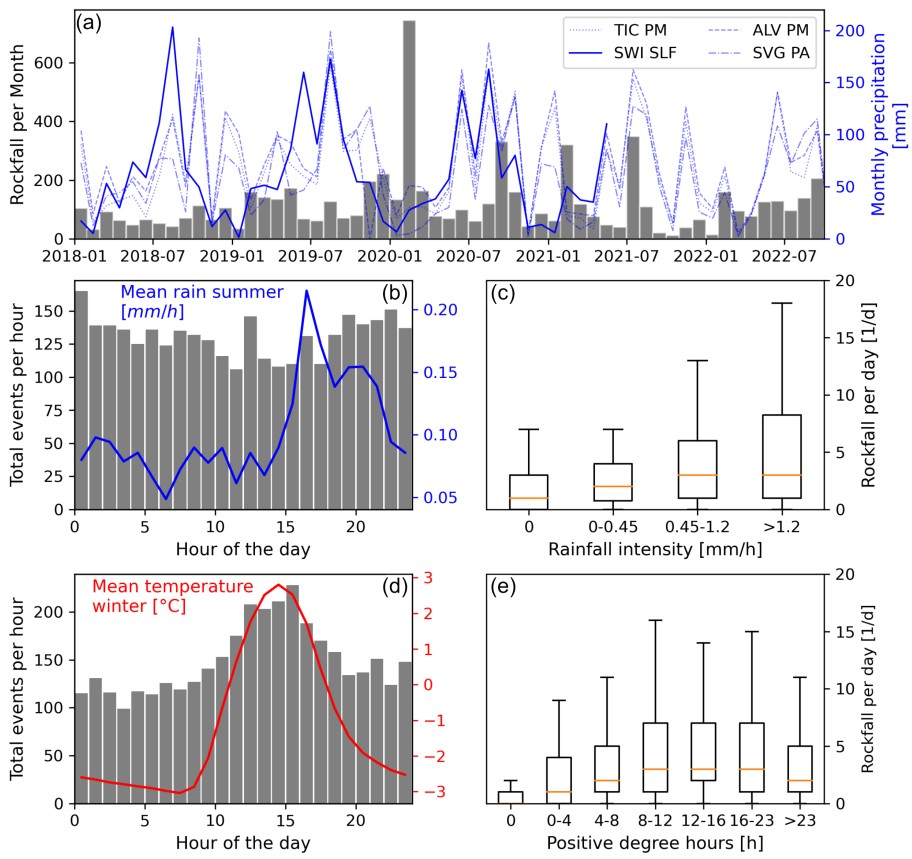

**Figure 10. (a)** Monthly Rockfall and Precipitation from Station Tiefencastel, Savognin, and Alvaneu. The wider blue line shows the SWI from SLF (2022). **(b)** Number of rockfalls for each hour of the day and in blue the hourly mean rainfall during summer (Station SVG PA). **(c)** Daily rockfalls as a factor of rainfall intensity (Station SVG PA) **(d)** Number of rockfalls for each hour and in red the hourly mean temperature during the winter months. Temperature from local station BRINZ **(e)** Daily rockfalls as a factor of duration of positive temperature during winter. Temperature also from local station BRINZ

Fig.10a shows monthly rockfall together with precipitation from Stations Savognin, Alvaneu, and Tiefencastel and the modeled SWI from SLF (2022). The Pearson correlation is generally low for both daily surface water input and temperature





(Tab. 2). It is assumed that only rain is precipitating during summer. The correlation analysis highlights a Pearson's correlation
coefficient of 0.33 between daily rockfall and rainfall during summer, while the correlation between daily water input and
rockfall is lower in winter. The effect of temperature on rockfall activity was also investigated using daily mean temperatures.
We find a low positive correlation in the winter months and a low negative correlation with mean/maximum temperatures in
summer (Tab. 2).

On the other hand, Fig. 10 shows that at hourly timescales, the meteorological parameters can have a stronger impact on
the rockfall activity. First, we examine the relationship between rockfall and precipitation during the summer season, shown
in Fig. 10b and c. Since snow can be neglected during summer, we assume that all precipitation falls as rain. Summer rainfall
often occurs in the late afternoon - early evening (see Fig. 10b), but most rockfall events take place during the night (Fig. 10b),
statistically corresponding to times with the most ground wetness. In addition, Fig. 10c exhibits a positive correlation between
the median hourly rainfall intensity (average per day) and the number of rockfall events in a day, suggesting that rockfall
activity increases during days with intense rainfall.

In Fig. 10d and e, we investigate short-term climatic triggers in winter. We find that both the rockfall activity and air
temperature have a significant daily cyclicity, with most rockfall events occurring in the daytime and fewer at night and early
morning hours. Similarly, the air temperature reaches its maximum in the early afternoon and its minimum in the morning.
Thus, the temperature curve follows the rockfall activity, with a Person's correlation coefficient of 0.94. The time with positive
temperature per day is compared with the rockfall activity (Fig. 10e) in winter and shows that the rockfall activity is higher on
days with potential freeze-thaw cycles and lower on days with only negative or only positive temperatures.

To further investigate the influence of freeze/thaw cycles, we calculated the freezing potential as defined by (D'Amato et al.,
2016) from daily mean temperatures measured at the local weather station. This further allows us to distinguish between
different phases of ice formation, expansion and melting during a freeze-thaw cycle. We observe that rockfall activity is lower
and even partially absent during negative cooling phases (Fig. 11), with a mean of 3.6 events per day. During negative warming,
rockfall activity increases slightly compared to the previous negative cooling phase, but still only 3.7 events per day happen on
average. In contrast, thawing phases lead to a clear increase in rockfall activity, with a mean of 6.0 events per day. The increase
in rockfall activity takes place mainly during the early phase of thawing.

## 5   Discussion

### 5.1   Rockfall Drivers and Triggers

A unique advantage of the Doppler radar monitoring technology compared to classical methods, such as periodic measurements
of deposited mass or cliff monitoring with laser scanners or structure-from-motion photogrammetry, is the very high and
reliable temporal resolution of rockfall activity. In fact, the methodology we have applied represents the first truly continuous
survey. This allows to get better insights into rockfall causal factors. Here we use the definitions for rockfall causal factors
described in Popescu (1994) and Loew et al. (2022): Preparatory causal factors are ground conditions and processes that
reduce the stability of potentially unstable rock compartments until failure occurs. When the time of failure can be associated

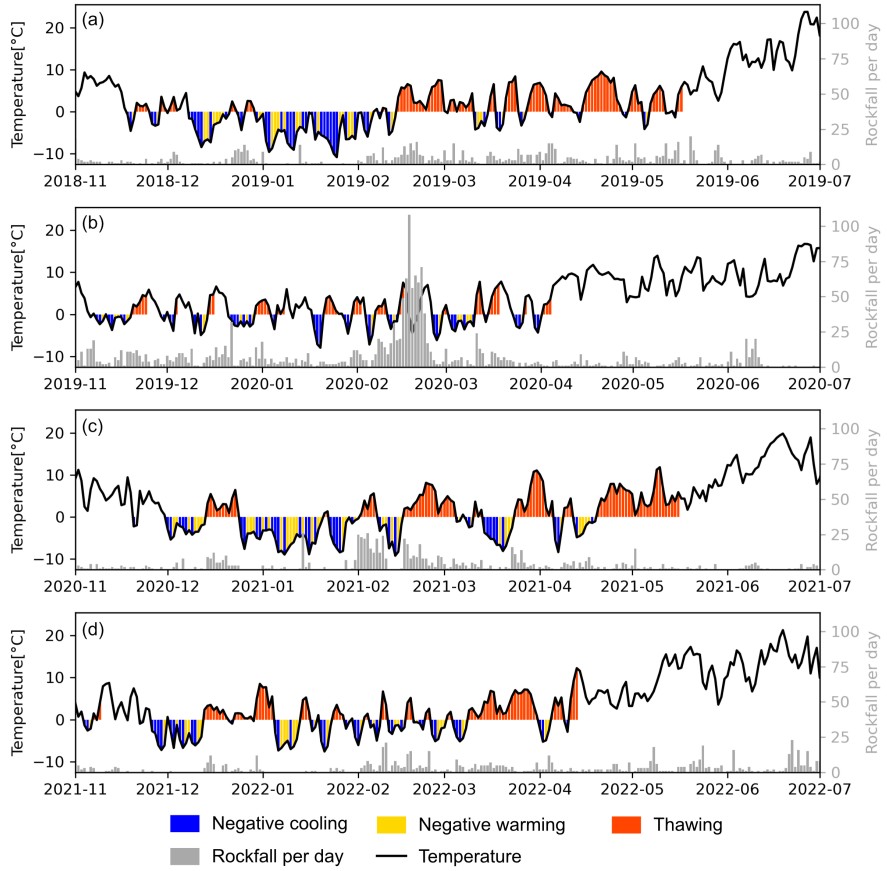

**Figure 11.** Temperature (black line) and rockfall activity (gray bars) in Brienz/Brinzauls for each freezing season from 2018 to 2022 (a, b, c, and d). Temperature is colored by the corresponding freezing potential phase. Blue indicates negative cooling, yellow for negative warming and red for thawing as defined by (D'Amato et al., 2016).

with an acting process, we call this a triggering causal factor. In past studies (D'Amato et al., 2016) about 50% of all rockfall events could not be related to a trigger. Most processes can both be preparatory and triggering causal factors.

In our rockfall release area, active rock sliding and rock toppling within a larger mountain slope deformation control high

displacement rates and are suggested to be the main preparatory causal factor. This is based on the following arguments: 1) Rockfall release areas correspond with locations of increased slope velocity as derived from terrestrial radar interferometry. Since the beginning of the terrestrial monitoring (starting in April 2019), different extraordinary active sub-compartments occurred within the "Insel" compartment and always correlated with areas of increased rockfall activity. This is also the main cause of the long-term motion of rockfall activity seen in Fig. 5. 2) Temporary clustered rockfall events over periods of several

weeks have not been observed in stable rock slopes (e.g., D'Amato et al., 2016) and can be clearly correlated with unstable compartments. This significantly impacts the short-term prediction of rockfall frequency and rockfall hazard management,





which for stable slopes can be correlated with daily weather conditions. The occurrence of clustered rockfall events is mainly driven by local rock mass damage and block rotation in unstable compartments and less by short-term climatic events (Fig. 6).

We observed that during the larger rock mass failure in February 2020, where $7500m^3$ of rock were released in several
hundreds of individual rockfalls (Loew et al., 2022), rockfall activity increased exponentially until a peak was reached on 18. February (Fig. 8). The following decrease in activity is in line with the deceleration of the remaining rock mass. We observed that most rockfalls during this time period originated in the area of the destabilized block (Fig. 9) and were more scattered later. Such an increase in rockfall activity prior to a larger mass failure is known as precursory rockfall and well described by Rosser et al. (2007) in Jurassic shale and sandstone cliffs of the coast of the North York Moors National Park, UK. They assume that
slope displacements lead to increased strain and fracturing of the rock, preparing slope sectors for rockfalls. Increased rockfall activity during the acceleration of unstable rockslides has been observed at other locations in crystalline and sedimentary rocks (Sartori et al., 2003; Oppikofer et al., 2008; Gschwind and Loew, 2018). However, the sites are located at different altitudes and exposed to different climatic conditions and rockfall drivers. Therefore, local rock slope displacement and acceleration can be seen as major rockfall preparatory factors. On the other hand, there exists no linear correlation between the mean annual
slope velocity of nearby monitoring points (Fig. 2) and rockfall frequency (Fig. 4).

Related to this causal factor, rockfall frequency in our study area is very high. Assuming that the majority of recorded events have a volume of a few $m^3$, we can estimate the activity parameter $A(m^3)$ for the total release area of about $100'000m^2$ as ranging between 78 and 190 $yr^{-1}\ hm^{-2}$. This is two orders of magnitude bigger than the reported activity parameters from stable slopes in continental cliffs (Hantz et al., 2020).

Weather-related factors such as precipitation and temperature are typically playing an important role as rockfall triggers in stable slopes (e.g., Loew et al., 2022). At Brienz/Brinzauls a simple correlation between rockfall and SWI is not evident. SWI and temperature both have annual cycles leading to seasonal rockfall patterns. Stronger seasonality in rockfall activity was recorded from 2020, starting with the February 2020 event cluster (Fig. 5), until 2021 and the seasonality was weaker in 2018, 2019, and 2022. Such changes in patterns are not uncommon and observed at other sites as well (Luckman, 1976; Gardner,
355 1971).

During summer, we found that rockfall activity per day is increased on days with higher rainfall intensity (Fig. 10c). At other sites, similar findings were presented by Krautblatter and Moser (2009), where rockfall activity increased non-linearly with rainfall intensity. However, thunderstorms may lead to very local rainout and, therefore, to an under- or overestimation of the actual effect on rockfall activity. Furthermore, the influence of precipitation on rockfall activity could also depend on the
source area location.

During winter, rockfall is most frequent during mid-day, which is in line with the hourly mean temperature (Fig. 8d). Ice can act as a cohesive force for rock blocks and increased rockfall activity is mainly observed during melting or thermal ice expansion periods above $-5°C$ (D'Amato et al., 2016). This could explain why fewer rockfalls are observed at night and at days with temperatures constantly below or above the freezing point (Fig. 10d,e and Fig. 11). In addition, meltwater infiltration
may lead to an increase in rockfall activity, as observed by Mourey et al. (2022) in the Mt. Blanc region. During the melting hours, rockfall frequency in our study area increases by a factor of about 2, with no observed time lag. Therefore, loss of



cohesion from melting ice infills in fractures is assumed to be the main cause for rockfall frequency increase during thawing periods.

Although the thawing phase can also generate meltwater infiltration, distributed fluid pressure increase in the shallow rockfall
release areas is very unlikely due to the high permeability and very low groundwater table of the rock mass. On the other hand, snowmelt and presumably meltwater infiltration in spring correlate with rapid accelerations of the "Insel" rockslide compartment.

## 5.2   Rockfall transport in the shadow zone

Assuming that a single block can be correlated with a Doppler radar event, about 3% of all events reach the shadow zone
of Brienz/Brinzauls. The shadow angle which has developed in the last 20y of increased rockfall activity is estimated to be approximately 27° (Fig. 7). Evans and Hungr (1993) evaluated the shadow angle of 16 rockfall talus profiles and found that 27.5° is a commonly observed shadow angle. The occurrence of a lower shadow angle, hence an increased reaching distance, was only observed in specific conditions such as smooth substrates covered with grass or snow. The underlying mechanism of rockfall transport in the shadow zone is mainly rolling blocks, and the extent of rolling is mainly controlled by the rolling
friction angle, which is not only influenced by the block sizes on the talus cone, but also by the substrate in the shadow zone and the properties of the man-made rockfall dam. The rockfall activity in Brienz/Brinzauls has led to the dam reaching its maximum capacity, as indicated in Fig. 2f. In the course of our investigations, we also revealed that compared to the years prior 2020, an increased number of smaller blocks reach the shadow zone directly (Fig. 2e). This increase in events is likely related to the flattening of the dam, as well as accompanying changes in the type of terrain.
Excavating the dam in the danger zone under the current very high activity rates is practically impossible. As a result, it is expected that the frequency of rockfall events reaching the danger zone will increase and smaller shadow angles could develop. Based on literature values, these could become as small as 24 to 25° but are not expected to endanger the houses in close proximity to the shadow zone.

## 5.3   Strength and limitations of a Doppler radar rockfall monitoring system

Rockfall monitoring methods have seen big technological advances in recent years, mainly related to developing high-resolution structure-from-motion photogrammetry, digital image correlation, and LiDAR investigations. These technologies can be applied from the ground, drones and piloted aircraft. Most investigations have been periodic and processing is not fully automated. In addition, optical methods only work under cloud-free and illuminated conditions. The Doppler radar offers continuous rockfall recording, also at night and under bad weather conditions (when many rockfalls are released). The time-lapse of events
recorded at Brienz/Brinzauls is in the range of up to a few minutes, as every larger rockfall event leads to automatic road closure and reopening. A higher recording frequency would be technically feasible. This high frequency and reliability under all weather and illumination conditions result in a unique time series of rockfall events above the critical size thresholds described in sect. 3.2. Such rockfall time series can be used to unravel short-term triggers in great detail. Additional and more detailed investigations of triggering processes using local climatic sensors and high-resolution cameras are underway.



The main objective of the Doppler radar at Brienz/Brinzauls is the automatic closure of the cantonal road by a traffic light within a few seconds after the detection of a rockfall event. After regular events where the blocks are deposited only on the talus cone, the road is automatically reopened after a few minutes. For large or clustered events reaching and traveling through the danger zone, the responsible risk management experts receive an automatic alarm via cellular network and the road is automatically reopened again after a few minutes. However, a yellow flashing light at the traffic lights will signal to the road

users that they must expect obstacles on the road. The experts then consult the radar data and optical images of the event taken at an interval of 3 seconds, enabling the operators to check the road for damages and depositions quickly. After this check the traffic light is reinstated. It is, therefore, essential that the system operates quickly and with a very low false negative rate, to ensure the safety of road users. During the entire period of operation, the system worked very reliably and no injuries were reported in this section of the road due to rockfall. The system allows the access road to Brienz/Brinzauls to stay at its current

location despite the very frequent events and high rockfall hazards.

For scientific data acquisition and transport process investigations, also detailed and accurate spatial information is required, including release, transport, fragmentation and deposition. However, the calculated intensity maps supplied by the Doppler radar are very blurred. The horizontal and vertical error is estimated to be at about $10m$ and $2°$ to $3°$, respectively. This makes a start point calculation difficult. Also, the transport path and runout distance are not well represented by the radar records, and

the time-lapse optical images must be consulted for detailed information. However, this does not affect the alarm. Furthermore, obtaining a sufficiently accurate volume estimation from the radar image is impossible. Such information would be valuable to determine the maximum block sizes that could potentially enter the zone of the road and village. The Doppler radar is, therefore, more suitable for analyzing rockfall events in time than their volume and detailed spatial extent.

## 6    Conclusions

We have investigated a new rockfall data set recorded by a Doppler radar at the active landslide complex of Brienz/Brinzauls in the Swiss Alps. This is the first complete rockfall time-series recorded during 4 years for events of about $1m^3$ size and larger. The rockfall release areas are mainly located in a south-viewing dolomite cliff of a very active landslide compartment (1500 to 1700 m a.s.l.), moving with more than $10m$ per year in downslope direction. We can show that under these active landsliding conditions, rockfall frequency defined by the activity parameter is two orders of magnitude higher than in stable continental

cliffs. While rockfall events are also reliably recorded during nights and bad weather conditions, monitoring transport pathways is less complete than other investigation methods. Nevertheless, the rockfall radar at Brienz could be used as a reliable automatic early warning system since 2018, closing an important cantonal road in a hazard zone that experienced 166 hits during the study period. This hazard zone is located in the rockfall shadow area and assuming that a single block can be attributed to a Doppler radar event, about 3% of all events reach the shadow zone of Brienz/Brinzauls. The shadow angle which has developed in the

last 20 years of increased rockfall activity is estimated to be approximately $27°$. We observe a frequency increase of rockfall events reaching the shadow zone, which can be attributed to the filling of a rockfall dam.



In our study area rockfalls occur in clusters lasting several days to weeks. These clustered events show very low correlations with daily weather conditions, considering rainfall, snowmelt, temperature and freeze-thaw cycles. On the other hand, the clustered rockfall events clearly correlate with the velocity of landslide activity hotspots in the highly fractured sliding and

toppling rock masses. These hotspots can be reliably detected and monitored by continuous ground-based radar-interferometry, and serve as a better early-warning indicator than weather data. On an hourly time scale, daily temperature maxima correlate with highest rockfall frequencies in winter. They indicate that warming and thawing periods of freezing ground are important short-term rockfall drivers during winter. Additionally, rockfall activity during the thawing periods is about twice as high as average. Due to the short instantaneous reaction of the rockfall activity, a direct connection to local rock mass acceleration due

to meltwater infiltration is assumed.

*Data availability.* The daily rockfall counts, spatial rockfall and freezing potential data sets are made available after publication at doi.org/10.3929/ethz-b-000605062. Up to this point, the data are under embargo.

*Author contributions.* All authors contributed to the study's conception and design. Data analysis was performed by all authors. The first draft of the manuscript was written by Marius Schneider and Nicolas Oestreicher and Simon Loew edited the document. All authors commented

on previous versions of the manuscript, read and approved the final manuscript.

*Competing interests.* The authors declare that no funds, grants, or other support were received during the preparation of this manuscript. The authors have no relevant financial or non-financial interests to disclose.

*Acknowledgements.* The authors would like to acknowledge the valuable contribution of the rockfall data set from Geopreavent AG (Thomas Ehrat), the landslide velocity data provided by CSD Ingenieure AG (Stefan Schneider) and the advanced surface water input model data by

SLF Davos (Rebecca Mott-Grünewald). Further we would like to thank the Swiss Seismological Survey SED for providing local weather data. We would also like to acknowledge the great help and support received from Amt für Wald und Naturgefahren Graubünden (Andreas Huwiler and Andri Largiadèr), Tiefbauamt Graubünden (Christoph Nänni) and BTG Büro für technische Geologie (Reto Thöny) who have provided reports, guidance and encouragement throughout the research.





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
