# Peer review of "Rockfall monitoring with a Doppler radar on an active rock slide complex in Brienz/Brinzauls (Switzerland)"

_EGUsphere, 2023_

## Author Comment (AC1)

Dear Anonymous Reviewer #1,

The authors would like to express their gratitude for your comprehensive comments. We acknowledge the feedback regarding writing style, table formatting, and figure layout, which will be applied after the comment of Anonymous Reviewer #2. We would like to provide a brief response to specific content-related comments:

1. Terminology: Danger Zone and Shadow Zone: We agree that the usage of these terms lacks clear definitions and coherence. Therefore, we also find it necessary to revise the use of these terms and provide a clear definition.

2. The local weather station used (l. 89-90) is a relatively cheep Holfuy-Station installed 2m above ground. The temperature senor has a radiation shield. Details are given in https://holfuy.com/de.

3. Temperature correction (l. 113): We apologize for the inclusion of this text sequence from an earlier draft. No temperature correction was applied to the local weather data.

4. Snow depth elevation correction (l. 119): The data utilized in this study originated from an unpublished study conducted by the SLF. The snow depth measurements were obtained through extensive LIDAR measurements using helicopter surveys, where in the end an accumulation field could be calculated. Hence, a very detailed actual snow depth distribution was received for the used time period.

5. Methods by Geopraevent AG (l. 146-150): The methods employed by Geopraevent AG are not intended for public disclosure and are subject to confidentiality agreements. As a result, we are not authorized to publish sensitive data or descriptions. We will provide a more detailed explanation of the public disclosure in the text to ensure complete transparency.

6. Activity parameter A (l. 346-349): The frequency analyses can display the cumulative or the non-cumulative distribution of the rockfall volumes. These distributions are usually fitted by a power law for the volume range where the inventory is exhaustive (e.g. Gardner, 1970; Hungr et al., 1999). Then the spatio-temporal frequency F of rockfalls bigger than a volume V can be expressed as:

$$F = A(V_0)(\frac{V}{V_0})^{-B} \qquad (1)$$

Where A is the frequency of rockfalls with a volume bigger than $V_0$ (an activity parameter) and B is a uniformity coefficient, which reflects the decrease of the frequency when the volume increases. $V_0$ is the minimal value of the considered volume range or a minimal volume of interest, which depends on the context of the analysis. See Loew et al. 2021.

Thank you for your valuable feedback, which will significantly contribute to improving the clarity and accuracy of our manuscript.

Sincerely,
The Authors

---

## Author Response (AR1)

**Response to Review Comments**

*We thank both reviewers for their careful reading and detailed comment. Most of these comments have been considered for the revision of the manuscript. Review comments from the reviewers are stated first followed by our "Replies". Note that the line and page numbers referred to in this response document correspond to the uploaded revised manuscript, in which the added/changed parts are highlighted in red.*

**Editor:**

Thank you for answering the comments of the reviewers. Your manuscript will be reviewed again after your implementation of the changes. As you are using data from GEOPRAEVENT without a co-author affiliated to them, it would be important for me to have a written statement of GEOPRAEVENT that you are allowed to publish their data.

> **Reply:** Good Suggestion. We do not publish the raw data from Geopreavent, as regulated in our contract. But the raw data have been essential for this work. Therefore, we have offered co-authorship to Geopraevent and they proposed to add Thomas Ehrat to the authors list, what we have implemented.

**Anonymous Referee #1:**

**General Comments**

While the data is worth publishing, this current manuscript needs a revision aiming for more consistent terminology (e.g., shadow zone, danger zone hazard zone) and concise writing with an enhanced reading flow (e.g., without "on the other hand"). As a non-native English speaker, I did not notice many linguistic errors. Instead, minor repetitions or the lack of an introduction (e.g., the activity parameter) should be tackled.

> **Reply:** The manuscript has been updated to make the terminology consistent and improve the reading flow.

**Specific Comments**

Line 3 (and throughout the whole manuscript): Spaces must be included between number and unit, without italics:1 m³, not *1m³*

> **Reply:** The manuscript has been updated to adhere to the prescribed formatting standards in terms of numbers and units.

Line 17-20: Although the reference is given to Evans and Hungr (1993), and the shadow zone of the rockfall process is described in detail with different wording, it might make sense to reference Figure 7, introducing a visual explanation of the shadow zone.

> **Reply:** We added a reference to figure 1d, which serves as a visual explanation of the shadow area.

Line 48 (47): Consider using Geopraevent AG or Geopraevent Ltd. to clarify their requirements as a private sector company (see methods).

**Reply:** We have added the abbreviation "AG" to address the proper company name of Geopraevent AG.

Line 50 (49): The reference is missing any DOI/internet address.

**Reply:** We have updated the reference to a newer version of the handbook and added the corresponding internet address to the reference: "Geopraevent: Monitoring systems for gravitative natural hazards, Tech. rep., Geopraevent, Zurich, https://www.geopraevent.ch/wp-500 content/uploads/2022/09/Technology_Guide_Geopraevent_2022_EN-1.pdf, 2022."

Figure 1: The layout of Fig. 1 could be enhanced: The inset map of Switzerland has much-unused space, but the legend and the main figure are hardly readable (in original zoom size). The information content of the figure can be increased if the weather-stations of Tab. 1 are additionally shown (certainly those of Brienz in the main map, but also the others in a larger section of the map or a second inset map of the region/canton). Also, displaying the longitudinal profile of Fig. 7, including the APEX point and the (additional) compartment names, would increase the information content. The designation of the danger zone needs to be clarified: Does it refer to the shadow zone, or is it based on the danger map of the canton?

**Reply:** We have thoroughly revised Figure 1. As suggested, the legend has been enlarged, the Swiss map has been downsized, and the weather station BRINZ has been incorporated into the map. Furthermore, in accordance with the recommendation from Anonymous Referee 1, a second inset map of regional scale was created, showing the locations of the weather stations Tiefencastel, Alvaneu and Savognin. As proposed, we also added the longitudinal profile to Figure 1 and present the trace in the main map. We also agree on the comment regarding the compartment naming. The nomenclature and spatial extent of the compartments were significantly simplified in the first manuscript. However, based on the feedback received and the recent crisis of the Insel compartment, we have changed the spatial distribution and nomenclature of the compartments to align with the established designation. In the updated version, we exclusively use the term "shadow area," which is also depicted in the main map, replacing the danger zone.

Figure 2: Although this figure and its caption give a good overview of the study area, some details might be enhanced or clarified: 2.a) Consider including the Photo location in Fig. 1. Caption 2.b): In the text is also the road closing infrastructure mentioned (l. 74). The red light visible in Fig. 2.b) is thus worth mentioning. Additionally, Fig. 2.b, c) would benefit from an overlay with the geological formations and/or the landslide compartments. Fig. 2.e) Why is there a shift between the mapped rock outlines and the orthophoto? Fig. 2.f) Mark the dams better (The lowest, far western dam directly on the road is barely visible). Again, both terms are mentioned (which might be right): Shadow and danger zone. Make it more clear by mapping the shadow zone.

**Reply:** We have implemented most of the proposed changes. In Figure 2a, the compartments of the landslide have been included. The photo location is provided in the image description, referencing Figure 1c. In Figure 2b, the traffic signal has been marked, and a geological overlay has been inserted. In Figure 2c, due to the absence of clear reference points for geological contacts, we have decided against including a geological overlay. In Figure 2e, the shift resulting from slope

movement and imprecise GPS measurements, has been corrected manually. In Figure 2f, the dam and trench have been delineated and described using red outlines.

Table 1: Only here you make use of parentheses. In the other figures' labels, you use brackets around the units. To enhance consistency, I would use only parentheses (as in most NHESS publications) or adapt at least here the parentheses to brackets.
    **Reply:** The parentheses were changed to brackets.

Line 75-77 (74-76): The observed velocity changes of the reflectors between the seasons belong (besides a good explanation) to the result section.
    **Reply:** We prefer to keep this text in the introduction. The landslide velocity is a background information of this study and not a finding of us or this paper.

Line 79 – 81 (80): Again a new term: "hazard zone". Its definition is also not concise. Instead, rephrase it: "We define this grassland including the cantonal road as a danger zone (Fig. 1), due to the increased damage potential (traffic)." This is necessary, as the grassland itself barely accounts for a higher damage potential.
    **Reply:** We have decided not to use the term "hazard zone." In the revised version of the manuscript, only the term "shadow area" is employed, but still addressing the special interest for these events. We appreciate the suggested rephrasing, which we incorporate in a slightly modified form in the revised manuscript.

Line 82 (81): Source of the large volume events?
    **Reply:** We have added the corresponding reference in line 83: Krähenbühl, R. and Nänni, C.: Ist das Dorf Brienz-Brinzauls Bergsturz gefährdet?, Swiss Bull. angew. Geol., 22/2, 33–47, https://doi.org/10.5169/seals-738126, 2017

Line 86 (85): "…and a/the local station…"
    **Reply:** the pronoun has been added.

Line 90-93 (89-90): Which temperature sensor is used, and which accuracy according to the manufacturer? Is the housing ventilated? What is the ground material (indirect radiation)?
    **Reply:** In the revised manuscript, details regarding the specific location, soil characteristics, ventilation conditions, and the sensor accuracy provided by the manufacturer (with an internet reference) have been included. "The local weather station, situated on the compartment Plateau (Fig. 1a), measures the temperature 2 m above a forest-like surface. The temperature sensor is protected by a non-ventilated radiation shield and is exposed to the south, thus facing direct sunlight. According to the manufacturer's specifications, temperature measurements are conducted with an accuracy of 0.5 °C (Holfuy, 2023)."

Line 94-98 (91-95): For clarity, I propose to rewrite the section. The most obvious choice to use the temperature data is the local station, as it is the closest one. So I would argue the other way around. E.g., "The proximity to the process area makes it suitable to use the temperature data set from the local

station for further analysis. A high overall correlation (0.93) has been observed between the temperature of Savogingn and the local station (BRINZ). The slight differences might be due to topographic temperature effects, as the station in Savognin measures lower mean temperatures in winter (2 °C), but higher temperatures in summer (1 °C)."

**Reply:** We appreciate this comment and gratefully incorporate the provided suggestion.

Line 101-114 (96-100): Please rewrite for clarity: The sentence "Secondly, freeze-thaw (FT) cycles can be divided into three phases." has the potential for confusion. Consider rewriting: "Secondly, freeze-thaw (FT) cycles, which we divide into three phases". Thanks to the proposed phrasing, it is clear, that only two metrics (l. 96) are meant and not three. Also, equations 6-8 could be moved directly after l. 100 to enhance clarity.

**Reply:** We appreciate the proposed phrasing, which is implemented in the revised manuscript. We also moved equations 6, 7 and 8 to the proposed position.

Line 119 (113): How is the temperature corrected?

**Reply:** We apologize for the inclusion of this text sequence from an earlier draft. No elevation correction was applied to the local temperature data. Hence, we deleted this text sequence.

Line 121-125 (119): Is the snow height also elevation-corrected?

**Reply:** The snow height is modelled as elevation dependent, and calibrated with detailed measurements of local snow height data.

Line 132-134 (127-132): Shorten the section: Slightly repetition within

**Reply:** We shortened the section and excluded the slight repetition. "This phenomenon […], while the corresponding change in the frequency is known as the Doppler shift frequency. This allows to detect and track fast movements in the field of view, which is used in regular daylife."

Line 139 (137): Reconsider: "The device has a 90° horizontal field of view, from 302° - 32° azimuth."

**Reply:** This text sequence was revised according the proposed way.

Line 142-144 (140-142): Unclear relationship of the volume and distance: "Within a distance of 100 m the radar can detect moving masses larger than 0.1m3. Increasing the distance to 1km, a minimum volume of 1m3 is necessary for detection."

**Reply:** This text section was rephrased to clarify the relationship between distance and required volume for detection:" The minimum detectable volume for moving objects depends on their distance from the radar system. Within 100 m line of sight, the radar can detect moving masses larger than 0.1 m$^3$. If the distance between moving object and radar station is increased to 1 km, a moving volume of at least 1 m$^3$ is required to be detected (Geopraevent, 2022)."

Line 145-146 (142-143): The minimal velocity is not mentioned in the source (Gassner et al., 2022)

**Reply:** Unfortunately, an error in referencing has occurred. We have corrected the reference to: Meier, L., Jacquemart, M., Wahlen, S., and Blattmann, B.: Real-time rockfall detection with Doppler radars, in: Proc. 6th Interdisciplinary Workshop on Rockfall Protection, pp. 75–78, 2017.

Figure 4: Consider writing "true positive wrong extent (TPWE)" instead of "TPWE", as this abbreviation is not common and in the figure is (in the current typesetting) before the mention within the text

    **Reply:** We changed the figure caption as proposed.

Line 150-155 (146- 150): Also, the signal-to-noise ratio is not explained in detail in the source (Gassner et al., 2022), However, such deeper insights are very welcome. Especially as the further "advanced algorithms" are described rather mysteriously. What do they do? And how? If these algorithms are developed by the company Geopraevent AG and are not meant to be disclosed, that should be mentioned clearly.

    **Reply:** The source is intended to reference the 1% false positive rate, not the signal-to-noise ratio. To clarify this matter, slight adjustments to the phrasing have been made. Moreover, a new section of text has been added to inform readers about the confidentiality agreement exists between the ETH authors and Geopraevent AG, specifically concerning the methods utilized by Geopraevent AG:

    " At this point we have to stress that the methods employed by Geopreavent AG are not intended for public disclosure and are subject to confidentiality agreements. Therefore, sensitive descriptions and insights into the algorithm structure can not be provided here."

Line 155: The Nr. of detected rockfalls belongs to the results section. However, another total amount of detected rockfalls (5667) is given for a slightly different period. Clarify which period is more meaningful.

    **Reply:** We kept the complete time series in the text (The system detected 6743 single events (time stamps) from 8 January 2018 to 5 October 2022) but moved it to the results section (line 205).

Line 182-184 (178): Specify how you include the caution in treating the eastern starting point or move this sentence to the discussion.

    **Reply:** This sentence is moved to the discussion

Line 187: Typo: "in the danger"?

    **Reply:** We have corrected this typo and changed the term danger zone to shadow area.

Line 196-199 (192): Which is the minimum rock size?

    **Reply:** The minimum block size is in the range of few decimeter of diameter. We added this information to the text, with additional information about random error and truncation bias.

Figure 5: Fig. 5.a and Fig. 5.d look very alike. Therefore, I propose that the subplot titles are descriptive and highlight the difference: "All rockfall event time series…" vs. "Days with high rockfall activity (>10 events d-1)".

    **Reply:** We changed the subplot titles as proposed

Line 210 (204): To maintain a concise terminology, you might change grassland to shadow- or danger zones.

> **Reply:** We changed the term grassland to shadow area.

Line 215-216 (209): The Doppler radar only observed rare events released from colluvial deposits.

> **Reply:** We implemented a similar change: The Doppler radar only rarely recorded events from the colluvial deposits.

Line 217-218 (211): The small second peak in the N-S-histogram of Figure 6 (now 7) may also result from the different orientations of the map and the radar/mountain exposition. Try to rotate the map such that the histograms get even more meaningful.

> **Reply:** We have adapted the proposed change and rotated Figure 7 by -13° (aligned with radar view). We can still observe the second peak in the N/S histogram. Nonetheless, we appreciate the feedback and apply the suggestion as recommended.

Line 219 (212): If the daily rockfall rate strongly fluctuates, besides the mean, the standard deviation would also be of interest.

> **Reply:** The standard variation is now also included in the text (7.3)

Line 219, 238, 258 (213, 231, 251): "Significantly" in terms of a statistical test? Which one? Rephrase otherwise.

> **Reply:** We decided to delete the term "significant", because no statistical test was conducted and only give qualitative measures.

Line 225-226 (219): To reference all subplots separately, write "..hereafter considered as days with high rockfall activity (Fig. 5d)."

> **Reply:** We have incorporated this change as suggested.

Line 256 (249): Consistency: Grassland or shadow: If you use it as a synonym, clarify that earlier and use just one term here. Otherwise, focus on shadow.

> **Reply:** We decided to focus on the term shadow area and have implemented it in the text.

Line 290-291 (285): To prepare the reader that the division between summer and winter also resulted in different subplots, you might add the corresponding references: "…we divide the data set into summer (April to September, Fig 10.b-c) and winter (October to March, Fig. 10.d-e) months."

> **Reply:** We have incorporated this change as suggested.

Table 2: After looking at the data plotted in Fig. 10 b and d, I would assume a negative coefficient with mean hourly summer rain (highest rockfall activity during the night, while rainfall events have their peak in the afternoon) and a positive coefficient for mean winter temperature (rockfall and temperature have their peak in the afternoon). Explain in the table caption (and text) more proactive the differences compared to Fig. 10 (daily/hourly). Even better: integrate the hourly correlation coefficients as well into the table.

**Reply:** We have integrated the Pearson correlation coefficients of the hourly meteorological data into Table 2. Additionally, we have adjusted the caption to clarify that the right side displays the correlation between daily values, while the left side displays the correlation between hourly values.

Figure 10: Adapt the y-axis label of Fig.10.b and d with the corresponding season for clarity. Although the caption correctly describes the summer and winter months, the bar graphs (b, d) are confusing because they show different data, despite identical axis labels: E.g., "Tot. events summer (h-1)"

    **Reply:** We have incorporated these changes as suggested.

Line 311-312 (307-310): This topic has already been introduced in the methods section. Shorten this repetition.

    **Reply:** We excluded the sentence "This further allows us to distinguish between different phases of ice formation, expansion and melting during a freeze-thaw cycle." from this paragraph to reduce the repetition.

Line 321-324 (316-320): hard-to-read, long introduction sentence, followed by two very short ones. Rephrase to avoid imbalance and enhance readability.

    **Reply:** We rephrased this part to: "Compared to classical methods such as periodic measurements of deposited mass or cliff monitoring with laser scanners or structure-from-motion photogrammetry, Doppler radar monitoring technology offers a unique advantage: it provides a remarkably high and reliable temporal resolution of rockfall activity. In fact, the methodology we have applied represents the first truly continuous survey allowing better insights into rockfall causal factors."

Line 347-352 (344-345): After an interesting thought and a good argumentation(342-344), mental agility is required by the reader to understand also the third sentence, which begins with a typical sentence intro (However, therefore, on the other hand). The wrong references (Fig.3 instead of 2 and Fig. 5 instead of 4?) and the prior missing link between the two figures are additionally unhelpful. Try to combine these figures into one and/or add the Pearson's correlation coefficients in Tab. 2. Then, the results section will describe the missing linear correlation, and here, the reader would not be surprised. Additionally, a reference to Fig. 8 could underline the argumentation and the importance of InSAR observations. "Therefore, local rock slope displacement and acceleration can be seen as major rockfall preparatory factors (Fig. 8). This insight is only possible thanks to the spatial InSAR observation, as the single, nearby monitoring points (Fig. 3 or new combined fig.) and rockfall frequency (Fig. 5 or new combined fig.) have no linear correlation. "

    **Reply:** Thanks again. We changed the wrongly referenced figures in this section and appreciate the proposed rephrasing which was fully incorporated into the text. Since we added the Pearson correlation coefficient to Table 2, we do not see any need to create a new combined figure.

Line 353-357 (346-349): Elaborate the activity parameter A and its calculation base in more detail. Why is the assumption of a few m3 per event necessary, but appears not in the unit of A? What is the unit of A anyhow: per year per hour per square meter? As your findings are due to the observed high rate very relevant, they should be better introduced.

**Reply:** Frequency analyses aim to estimate the temporal or the spatio-temporal frequency of rockfalls as a function of their volume. The spatio-temporal frequency allows to compare the rockfall activities of different cliffs. The frequency analyses can display the cumulative or the non-cumulative distribution of the rockfall volumes. These distributions are usually fitted by a power law for the volume range where the inventory is exhaustive (e.g. Gardner, 1970; Hungr et al., 1999). Then the spatio-temporal frequency F of rockfalls bigger than a volume V can be expressed as:

$$F = A(V_0)(\frac{V}{V_0})^{-B}$$

Where A is the frequency of rockfalls with a volume bigger than $V_0$ (an activity parameter) and B is a uniformity coefficient, which reflects the decrease of the frequency when the volume increases. $V_0$ is the minimal value of the considered volume range or a minimal volume of interest, which depends on the context of the analysis. Hantz et al. (2020) analyzed the influence of ground conditions on the activity parameter $A(V_0)$, using topographical inventories from 13 French natural cliffs in different geological and erosional conditions. For continental cliffs under 2000 m in altitude, the activity parameter $A(1 \text{ m}^3)$ varies from 0.02 $\text{yr}^{-1}.\text{hm}^{-2}$ for a massive limestone cliff to 1.5 $\text{yr}^{-1}.\text{hm}^{-2}$ for a thinly bedded limestone cliff ($1 \text{ hm}^2 = 10^4 \text{ m}^2$). We have expanded the text accordingly.

Line 393-395 (383-384): Can this assumption ("is likely related") be underlined with data? Different DEM over time?

**Reply:** This statement is based in visual observations between 2018 to end of 2022. We have rephrased this sentence to emphasize that it is a hypothesis that should be examined in further studies.

Line 396-398: Inconsistency in the terminology: Does your mapped "Danger zone" (Fig. 1) contain the here mentioned dam? Even if so, the second sentence does not make sense: reaching the dam would then mean: reaching the danger zone.

**Reply:** After the redefinition of the terminology, we consider the dam not as a member of the shadow zone – rather as the upper boundary. The shadow area is reached if blocks travel over the dam. Therefore we deleted the term "danger zone" in this specific case.

Line 398-399 (387): If a literature study is mentioned, it would be helpful to provide the sources.

**Reply:** We added the reference "Evans, S. G. and Hungr, O.: The assessment of rockfall hazard at the base of talus slopes, Can. Geotech. J., 30, 620–636, https://doi.org/10.1139/t93-054, 1993" related to the statement about changes of the shadow angle.

Line 411-414: Twice "the road is automatically reopened after a few minutes". Rewrite for reading flow.

**Reply:** We have rewritten this paragraph for better reading flow.

Line 411-418 (402-405): Twice "the road is automatically reopened after a few minutes". Rewrite for reading flow.

**Reply:** We have rewritten this paragraph for better reading flow: "The main objective of the Doppler radar at Brienz/Brinzauls is the automatic closure of the main road by a traffic light within a few seconds after the detection of a rockfall event. After regular events where the blocks are deposited on the talus cone, the road is automatically reopened after a few minutes. For large or clustered events reaching over the retaining dam into the shadow area, the responsible risk management experts receive an automatic alarm via cellular network and the road is automatically reopened after data consultation on the data portal. Still, a yellow flashing light at the traffic lights will signal to the road users that they must expect obstacles on the road. The experts then consult the radar data and optical images of the event taken at an interval of 3 seconds, enabling the operators to check the road for damages and depositions quickly. After this check the traffic light is reinstated."

Line 410-422 (400-410): Own data? Otherwise: Sources?

**Reply:** The information regarding the traffic light operation was provided by S. Wahlen (Geopreavent AG) by personal communication at January 22, 2023. We added this source in the Manuscript, line 411. The statement regarding the number of injuries is based on own observations.

Line 432-443 (420-430): Mentioning the activity parameter and the dam filling in the conclusion stresses the importance of the comments Lines 346-349 and Lines 383-384.

**Reply:** We have adjusted the paragraphs about the activity parameter and dam filling.

Line 448-449 (436): Consider also including the limitations of the previously widely used monitoring system into the conclusion: "than weather data or single monitoring points."

**Reply:** We have incorporated these changes as suggested.

References:

Gassner, J., Wahlen, S., and Meier, L.: Radarüberwachung von Massenbewegungen, Schweizerische Zeitschrift für Forstwesen, 173, 124–129, https://doi.org/10.3188/szf.2022.0124, 2022.

Geopraevent: Monitoring systems for gravitative natural hazards, Tech. rep., Geopraevent, Zurich, https://www.geopraevent.ch/wp-500 content/uploads/2022/09/Technology_Guide_Geopraevent_2022_EN-1.pdf, 2022.

Holfuy: Technical info Holfuy weather stations, https://holfuy.com/de/about, 2023.

Krähenbühl, R. and Nänni, C.: Ist das Dorf Brienz-Brinzauls Bergsturz gefährdet?, Swiss Bull. angew. Geol., 22/2, 33–47, https://doi.org/10.5169/seals-738126, 2017

Meier, L., Jacquemart, M., Wahlen, S., and Blattmann, B.: Real-time rockfall detection with Doppler radars, in: Proc. 6th Interdisciplinary Workshop on Rockfall Protection, pp. 75–78, 2017.

**Anonymous Referee #2:**

**General Comments**

The title could be refined to better reflect the focus on analyzing rockfall triggering and evaluating rock arrest locations using Doppler Radar.

> **Reply:** In this publication, we present rockfall data derived from a Doppler radar that has been employed for the first time ever (worldwide to our knowledge) as a rockfall monitoring and automated early warning system. Based on these data, we subsequently study and discus the rockfall triggering factors and address the strengths and limitations of the system. Another important difference to other rockfall studies is that we study causal effects in an active rockslide and not a stable cliff. We consider these aspects as distinctive features of this publication, and therefore, we prefer not changing the title.

Additionally, the inclusion of a geological map would greatly enhance the understanding of the study, considering the frequent references to geological aspects throughout the text.

> **Reply:** Thanks. Beside the changes recommended by Anonymous Referee #1, we also included a geological map in Figure 1 (see Fig. 1b). Reference for the geological Data: BTG: Geologischer Synthesebericht - Bericht 5897-19, Tech. rep., BTG Büro für Technische Geologie AG, Sargans, 2022.

Section 3.3 requires a clearer explanation, potentially aided by a flowchart to elucidate the selection process for the displayed source areas in Figure 4. Further clarification is needed regarding the manual mapping of 70% of the 971 events on pictures and the utilization of machine learning, particularly since only 10% are allocated as test sets.

> **Reply:** We have included a flow chart regarding the start point calculation process, containing the ML cleaning, start point calculation, rotation and final data cleaning with ArcGIS Pro.

Additionally, the authors could consider utilizing cross-correlation analysis to examine the time delay between climate effects and rockfall triggering. Lastly, certain sections, such as lines 400 to 410, seem irrelevant to the paper's purpose.

> **Reply:** In the initial phase of the study, cross-correlations were also conducted between rockfall activity and meteorological factors. However, no significant time lag was observed in these analyses. Consequently, we focused on investigating the direct correlation between rockfall and environmental parameters. Regarding the last comment, we think the new Doppler radar methodology should also be critically discussed with respect to the method limitations.

In terms of references, while D'Amato et al. (2016) is cited for freezing and thawing, it is also relevant to rainfall effects.

> **Reply:** Thanks

The term "Cantonal Road" requires clarification, as it appears to be a Swiss classification that could be replaced with "main road" for better comprehension.

> **Reply:** Thanks, we have implemented this in the revised text.

Exploring the influence of temperature alone could provide valuable insights, especially considering the south-facing slope indicated in Figure 10d, where the effect of freezing and thawing cycles appears immediate, though the feasibility of such an analysis may be challenging.

> **Reply:** We think this has been addressed in Table 2 (average daily temperature) and Figure 10d (hourly temperature in winter) and in the corresponding text.

**Specific Comments**

Line 36-37 (37): Consider adding the following reference: Hungr, O., Evans, S.G., and Hazzard, J. 1999. The magnitude and frequency of rock falls and rockslides along the main transportation corridors of southwestern British Columbia. Canadian Geotechnical Journal, 36: 224–238. doi:10.1139/t98-106.

> **Reply:** We acknowledge this suggestion and have added the reference in the revised manuscript.

Figure 1: It is recommended to include a geological map alongside the existing figures.

> **Reply:** Thanks, see answer b in General Comments

Figure 3: Clarify the meaning of "TPS."

> **Reply:** TPS is the abbreviation for Theodolite based Positioning System. We have added this clarification to the caption of Figure 3.

Lines 209 and 215 (203 and 208): Specify whether 60 m3 and 100 m3 refer to maximum block volume, and source volume respectively.

> **Reply:** In the text, we distinguish between block volume and event volume. The latter describes the total volume of failed material, while the block volume describes the volume of a single block within the failed volume. We have added this information to clarify this distinction.

Line 220 (202): Provide information on the minimum and maximum per day values.

> **Reply:** The per day values range from 0 to about 100 (see Figure 8).

Figure 7 (Figure 6): Improve the legend's clarity by explaining the histogram and consider using lines instead of transparent histograms.

> **Reply:** Thanks. Beside the suggested changes of Anonymous Referee #1, we also changed the histogram from transparent to line style. We also added additional clarification in the legend of Figure 7.

Consider employing the term "shadow area" or "rockfall shadow" instead of "shadow zone," as previously used by Evans and Hungr.

> **Reply:** Thanks, we have fully implemented this comment in the revised text.